# Post-systolic shortening index by echocardiography evaluation of dyssynchrony in the non-dilated and hypertrophied left ventricle

Yoshihito Saijo[1☯], Tom Kai Ming Wang[1☯], Nicholas Chan[1], Brett W. Sperry[2], Dermot Phelan[3], Milind Y. Desai[1], Brian Griffin[1], Richard A. Grimm[1], Zoran B. Popović[1]*

1 Section of Cardiovascular Imaging, Heart, Vascular and Thoracic Institute, Cleveland Clinic, Cleveland, Ohio, United States of America, 2 Mid America Heart Institute, Saint Luke's Hospital, Kansas City, Missouri, United States of America, 3 Atrium Health Sanger Heart and Vascular Institute, Charlotte, North Caroline, United States of America

☯ These authors contributed equally to this work.
* popoviz@ccf.org

**Data Availability Statement:** All relevant data are within the paper and its Supporting Information files.

## Abstract

### Background

Post-systolic shortening index (PSI) is defined as myocardial shortening that occurs after aortic valve closure, and is an emerging measure of regional LV contractile dysfunction. PSI measurement variability amongst software vendor and its relationship with mechanical dyssynchrony and mechanical dispersion index (MDI) remains unknown. We evaluated PSI by speckle-tracking echocardiography from several vendors in patients with increased left ventricular wall thickness, and associations with MDI.

### Methods

This is a prospective cross-sectional study of 70 patients (36 hypertrophic cardiomyopathy [HCM], 18 cardiac amyloidosis and 16 healthy controls) undergoing clinically indicated echocardiography. PSI was measured using QLAB/aCMQ (Philips), QLAB/LV auto-trace (Philips), EchoPAC (GE), Velocity Vector Imaging (Siemens), and EchoInsight (EPSILON) software packages, and calculated as 100%×(post systolic strain–end-systole strain)/post systolic strain.

### Results

There was a significant difference in mean PSI among controls 2.1±0.6%, HCM 6.1±2.6% and cardiac amyloidosis 6.8±2.7% (p <0.001). Variations between software vendors were significant in patients with pathologic increases in LV wall thickness (for HCM p = 0.03, for amyloidosis p = 0.008), but not in controls (p = 0.11). Furthermore, there were moderate correlations between PSI and both MDI (r = 0.77) and left ventricular global longitudinal strain (r = 0.69).

**Funding:** The author(s) received no specific funding for this work.

**Competing interests:** The authors have declared that no competing interests exist.

## Conclusion

PSI was greater in HCM and cardiac amyloidosis patients than controls, and a valuable tool for dyssynchrony evaluation, with moderate correlations to MDI and strain. However, there were significant variations in PSI measurements by software vendor especially in patients with pathological increase in LV wall thickness, suggesting that separate vendor-specific thresholds for abnormal PSI are required.

## Introduction

Post-systolic shortening index (PSI) is a phenomenon of regional contraction occurring after end-systole, and is most commonly detected by speckle tracking echocardiography. PSI is measured as a ratio between amount of deformation (strain) occurring after end-systole divided by total deformation [1]. PSI is commonly used as a marker of ischemia, and can occur in segments of prior ischemic damage [2]. However, PSI is also intrinsically related to mechanical dyssynchrony. The presence of post systolic shortening by definition means an increase in mechanical dispersion index (MDI), a dyssynchrony measurement that represents the standard deviation of time-to-peak of segmental strains. On the other hand, increased MDI also means the presence of PSI because contraction of some segments must have occurred after end-systole.

In a normal heart, contraction of the left ventricle (LV) is geographically coordinated so that fiber shortening in a given muscle wall layer occurs almost synchronously and by a similar amount throughout the chamber. In contrast, normal contraction pattern can be disrupted by LV hypertrophy even in the absence of overt conduction disease, resulting in a dyssynchronous contraction pattern [3]. Indeed, we have recently shown that MDI is increased in patients with increased LV wall thickness due to HCM or amyloid heart disease [4]. Even so, several studies have shown that vendor and software variability and measurement accuracy can influence speckle-tracking echocardiography derived parameters, and is more apparent when dealing with regional data [1, 5, 6]. Our study group reported that the confidence intervals for the agreement in MDI measurements amongst software vendors are large [4]. However, the agreement of PSI measurement variability among software has not been previously evaluated.

The aims of the present study are to evaluate speckle-tracking echocardiography derived PSI in patients with a pathological increase in LV wall thickness, assess the influence of vendor on post systolic shortening parameters, and analyze the relationship between PSI and mechanical dyssynchrony.

## Materials and methods

### Study population

Patient characteristics for this study have been previously reported [6]. In brief, consecutive patients at the Cleveland Clinic who underwent a clinically indicated outpatient echocardiography examinations between June 2016 and June 2017 were included if they had a diagnosis of hypertrophic cardiomyopathy or cardiac amyloidosis defined previously [6]. All HCM and amyloidosis patients were symptomatic. Amongst HCM patients, diagnosis was confirmed by cardiac magnetic resonance in 29 [7], and by subsequent operative histology findings in 3 patients, while the remaining were diagnosed by echocardiography alone (3 because of having an implantable cardiac defibrillator). Eleven patient underwent genetic testing, with

pathogenic mutation present in 7 subjects [8]. In addition, we recruited, as a control group, healthy subjects without valvular, structural, or coronary artery disease and normal systolic and diastolic function. Demographic and electrocardiogram characteristics were recorded, including QRS duration and the presence of left bundle brunch block. The study was approved by the Cleveland Clinic Institutional Review Board and Ethics Committee (number 16–724), and all patients provided written informed consent to be included.

## Echocardiography

Each patient underwent three scans using the GE Vivid7 or Vivid9 (GE Medical, Milwaukee), Acuson Sequoia SC2000 (Siemens Medical Solution USA, Inc., Malvern, Pennsylvania), and EPIQ 7C (Philips Medical Systems, N.A., Bothell, Washington) ultrasound scanning systems. A conventional echocardiographic study was performed firstly using a GE Medical machine. Subsequently, in the same imaging session and by the same sonographer, images were recorded using Siemens and Philips machines. Images were stored as raw data in a proprietary company format if available, and also in standard Digital Imaging and Communications in Medicine formats. Left ventricular (LV) end-diastolic volume, end-systolic volume, ejection fraction, left atrial volume indexed and average septal and lateral E/e' were recorded.

**Speckle-tracking strain analyses.** Three consecutive cardiac cycles of apical 4-, 2- and 3-chamber view images were stored for speckle tracking echocardiography-based strain analysis for all scanners under optimized machine settings and frame rates (>60/sec). Two-dimensional speckle-tracking analyses were performed using a total of five speckle-tracking software packages (Fig 1). Then, a first observer (YS) performed all MDI measurements, while a second observer (NC) performed all PSI measurements. Observers were blinded to each other's results. Off-line analysis of GE images was performed using the EchoPAC version 11.3 (GE

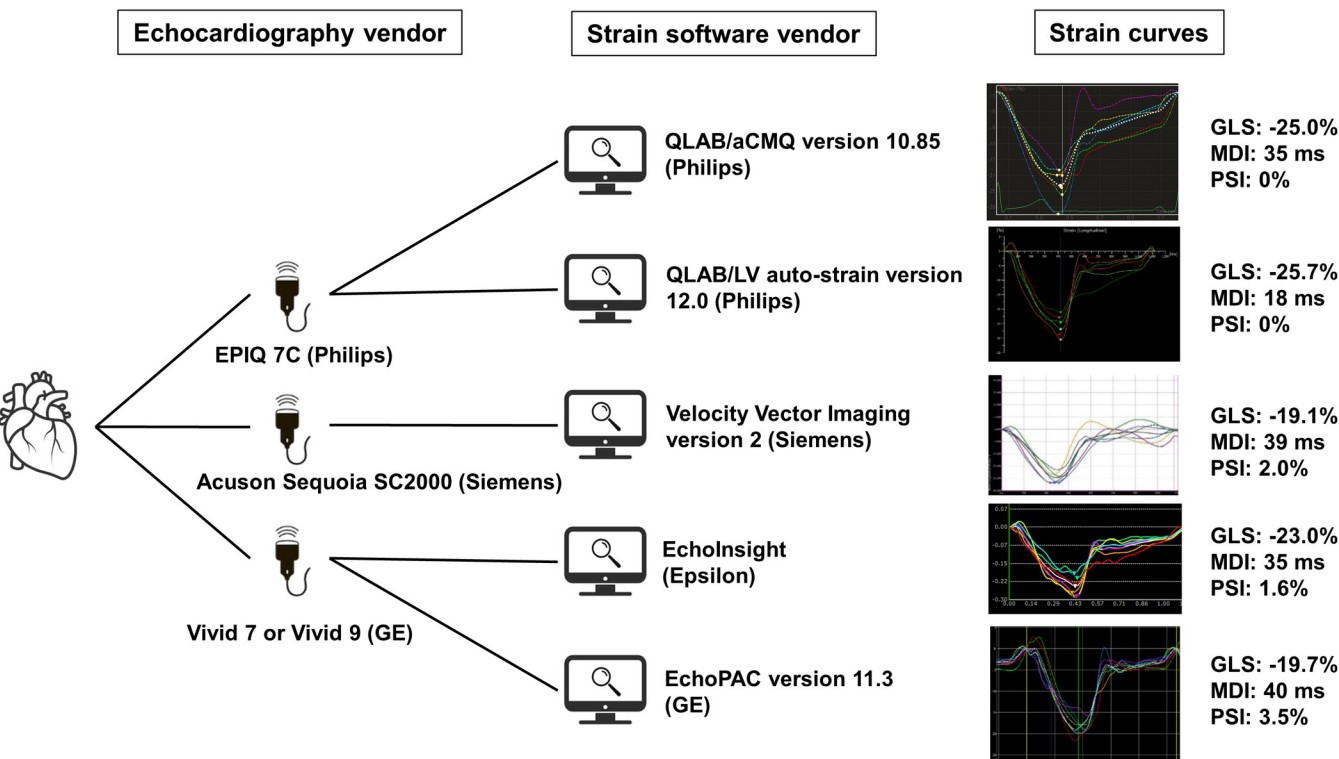

**Fig 1. Echocardiographic scanner and analytic software used in this study with example strain curves from one patient reporting global longitudinal strain (GLS), mechanical dyssynchrony index (MDI) and post-systolic shortening index (PSI).**

Medical System). Analysis of Philips images was performed using two different software packages: QLAB/automated cardiac motion quantification (aCMQ) version 10.85, and QLAB/LV auto-strain version 12.0 (Philips Healthcare). Analysis of Siemens images was performed using Velocity Vector Imaging (VVI) version 2 (Siemens Medical Solution). Additionally, we analyzed images from a GE Medical machine using a vendor-independent software program EchoInsight (Epislon imaging, Ann Arbor MI). The cardiac cycle with the best image quality was selected, and the endocardial border was traced manually or automatically. Subsequently, manual adjustments were made after visual inspection of the segmental tracking results through the cardiac cycle. The segments with adequate tracking were excluded from the analysis. Finally, the software automatically generated time-domain LV strain curves for each segment. Left ventricular global longitudinal strain (LVGLS) was calculated by averaging the negative peak strains of the 18 LV segments derived from 3 apical views.

**Post-systolic shortening index.** Timing of aortic valve closure was judged from apical 3ch-view. We calculated post-systolic shortening index (PSI) of each segment, defined as 100% × (maximum post-systolic strain–end-systole strain)/maximum post-systolic strain (Fig 2A) If the maximum longitudinal shortening was within the systole, PSI was set to zero. Global PSI was summarized and averaged to provide a mean value. Fig 3 shows examples of the PSI segmental plots for controls, HCM and amyloidosis patients.

**Mechanical dispersion index.** Time-to-peak negative interval was measured from the peak of the R wave of the ECG to the peak negative strain in each 18 LV segments, and MDI was calculated as the standard deviation of the 18 LV segmental time-to-peak intervals (Fig 2B).

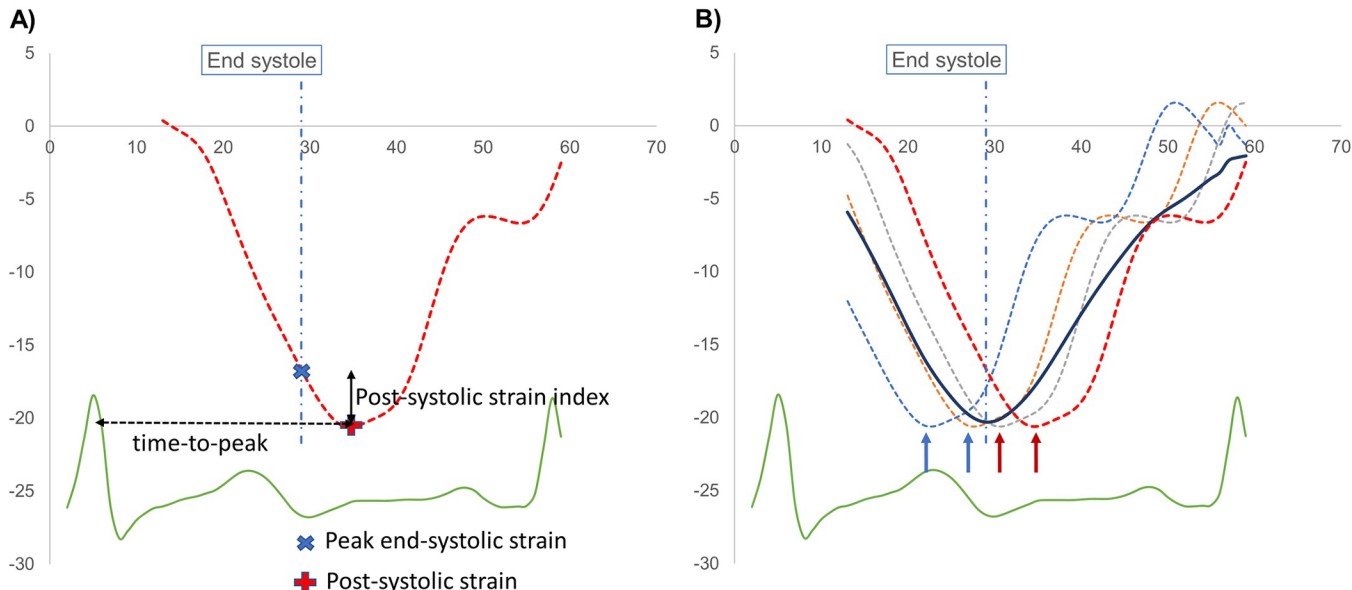

**Fig 2.** Schematic representation of Strain curves demonstrating (a) method of calculation of segmental post-systolic shortening index and time to peak strain and (b) relationship between global mechanical dispersion index and post systolic strain index. **Panel A** shows a single time-delayed segmental strain curve with values of peak end-systole strain (global) and maximum post-systolic strain marked by arrows. From these values, segmental post systolic strain index is calculated as: 100% × (maximum post systolic strain–end systole strain)/maximum post-systolic strain. If no regional post systolic strain present, its post-systolic strain index is 0. Time to peak interval, used to calculate mechanical dispersion index, and is represented by a broken arrow. **Panel B** shows four segmental strain curves dispersed in time and displayed as dashed lines, with their point-by-point average represented as a full black line. Two segmental strains peaking before end systole and having zero post systolic shortening are marked by blue arrows, while two segmental strains peaking after end systole and thus exhibiting post systolic shortening are marked by red arrows. Global post systolic shortening index is then calculated as the mean of segmental systolic shortening indices. Mechanical dispersion index is calculated as the standard deviation of segmental time-to-peak intervals. By default, absence of mechanical dispersion means absence of post-systolic shortening and vice versa.

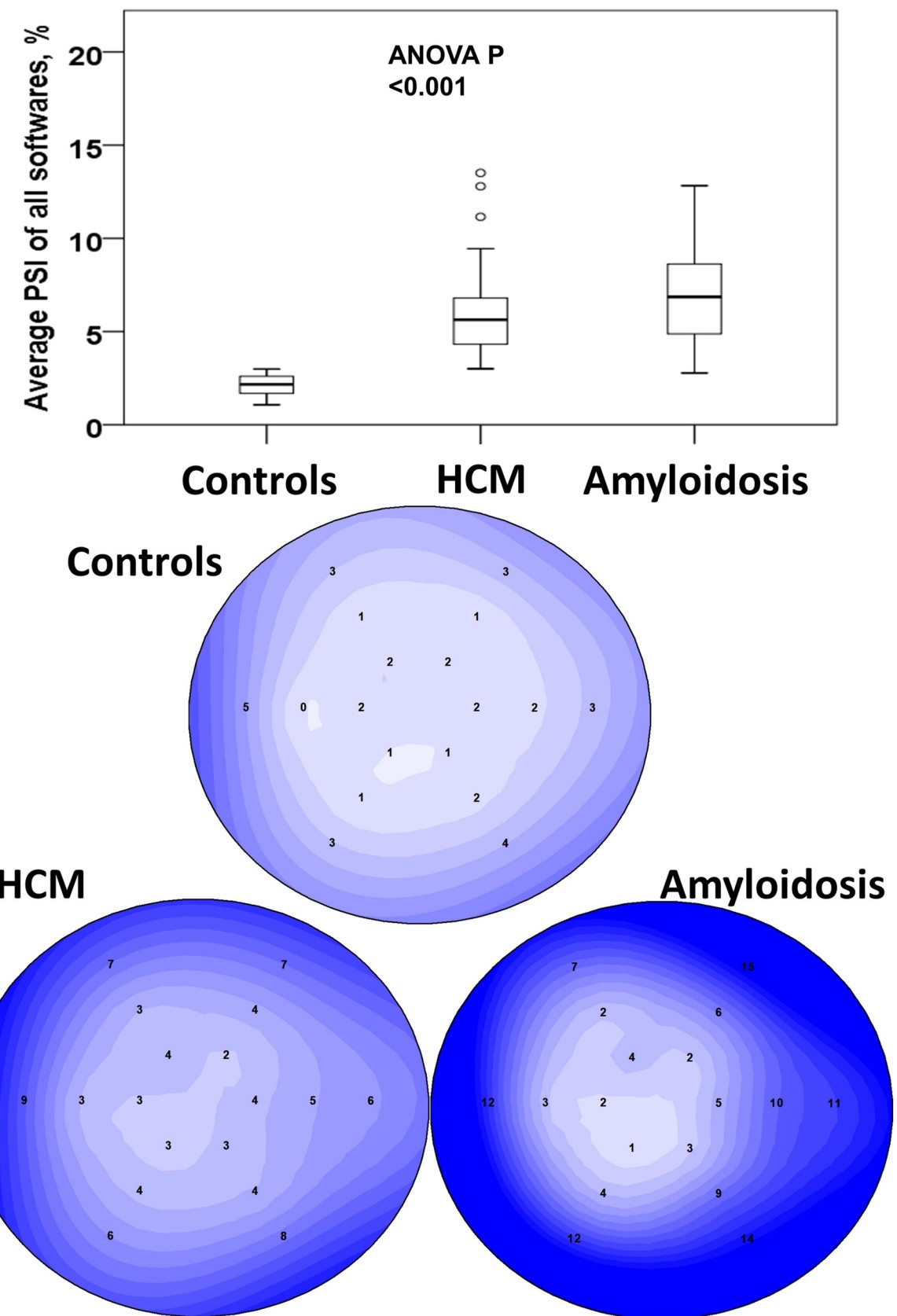

**Fig 3.** Comparison of post-systolic shortening index averaged across all strain software between controls, hypertrophic cardiomyopathy (HCM) and cardiac amyloidosis patients (top) and examples of post-systolic shortening index plots for healthy control, hypertrophic cardiomyopathy (HCM) and cardiac amyloidosis patients (bottom).

## Statistical analyses

Continuous variables were expressed as mean±standard deviation, and categorical data were presented using percentages and frequencies. Continuous variables were compared between the 3 patient groups by one-way analysis of variance (ANOVA), and among software vendors by repeated ANOVA measurement. Categorical data were compared among groups by chi-square test. Boxplots were used to illustrate mean and distribution of PSI between patient groups and strain software vendors. Pearson coefficients and scatterplots were used to examine the correlation between PSI and MDI, QRS duration and LVGLS. To assess the comparative accuracy of MDI and PSI measurements in identifying dyssynchrony, Receiver operating characteristics analyses were used to calculate the area under the curve for MDI and PSI to identify left bundle branch block and QRS duration $\geq$120 ms. P-values<0.05 were interpreted as indicating statistical significance and all tests were two-tailed. Statistical analysis was performed using IBM SPSS Statics version 25 (SPSS, Chicago, IL).

## Results

### Clinical and echocardiography characteristics

The study cohort included 70 patients: 36 patients with HCM, 18 with amyloidosis and 16 controls. Table 1 shows clinical characteristics and conventional echocardiographic measurements of these subjects, and the cohort's minimal dataset can be found in the S1 Table. Overall mean age was 57 ± 18 years, 42 (58%) were male, 6 (9%) had left bundle branch block, and 13 patients (19%) had QRS duration >120 ms. Between the three groups, cardiac amyloidosis had the highest age, proportion of men, heart rate, QRS duration, left atrial volume indexed, E/e', and lowest left ventricular ejection fraction, while controls had the lowest age, proportion of

**Table 1. Cohort clinical and echocardiography characteristics.**

|  | All | Control | HCM | Amyloidosis | P-value |
|---|---|---|---|---|---|
|  | (n = 70) | (n = 16) | (n = 36) | (n = 18) |  |
| Age (years) | 57 ± 18 | 43 ± 18 | 56 ± 17 | 71 ± 9 | <0.001 |
| Male | 42 (58) | 5 (31) | 23 (61) | 14 (78) | 0.021 |
| Heart rate (bpm) | 66 ± 13 | 69 ± 8 | 60 ± 11 | 76 ± 16 | <0.001 |
| QRS duration (ms) | 108 ± 32 | 93 ± 9 | 106 ± 24 | 123 ± 50 | 0.031 |
| Left bundle branch block | 6 (9) | 0 (0) | 6 (17) | 0 (0) | 0.049 |
| Implantable cardiac defibrillator | 12 (17) | 0 (0) | 7 (19) | 5 (28) | 0.087 |
| LV end-diastolic volume (mL) | 100 ± 31 | 105 ± 25 | 101 ± 34 | 93 ± 29 | 0.49 |
| LV end-systolic volume (mL) | 41 ± 15 | 42 ± 13 | 37 ± 14 | 48 ± 19 | 0.056 |
| LV ejection fraction (%) | 60 ± 9 | 60 ± 5 | 63 ± 6 | 50 ± 12 | <0.001 |
| LA volume indexed (ml/m$^2$) | 38 ± 13 | 25 ± 4 | 40 ± 12 | 43 ± 15 | <0.001 |
| E/e' | 13.7 ± 7.9 | 7.6 ± 2.2 | 14.1 ± 6.6 | 18.7 ± 10.3 | <0.001 |
| LV global longitudinal strain (%) | -17.0 ± 4.0 | -21.0 ± 1.4 | -17.6 ± 2.4 | -12.3 ± 3.8 | <0.001 |
| Mechanical dyssynchrony (ms) | 59 ± 19 | 38 ± 7 | 65 ± 15 | 64 ± 20 | <0.001 |
| Post-systolic shortening index (%) | 5.4 ± 4.2 | 2.1 ± 0.6 | 6.1 ± 2.6 | 6.8 ± 2.7 | <0.001 |

Values are mean ± standard deviation or frequency (%). HCM = hypertrophic cardiomyopathy, LV = left ventricular; LA = left atrial.

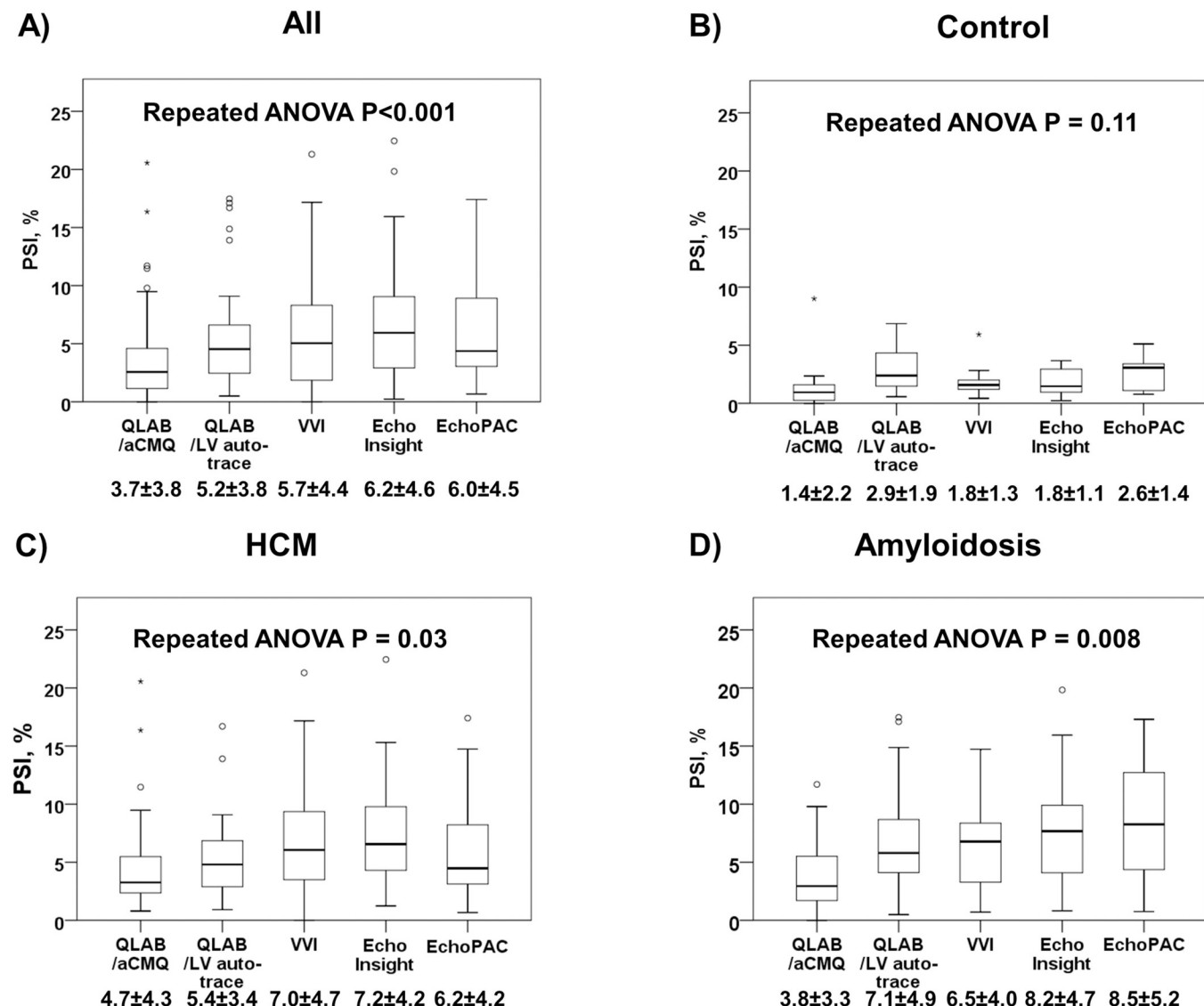

**Fig 4.** Post-systolic shortening index (PSI) by strain software in (a) all patients, (b) healthy controls, (c) hypertrophic cardiomyopathy (HCM), and (d) cardiac amyloidosis.

men, QRS duration, left atrial volume indexed, E/e'. All HCM and amyloidosis patients were symptomatic".

### Speckle tracking echocardiography assessment

Speckle-tracking images were recorded with mean frame rate of 85 ± 23 bpm for GE, 51 ± 3 bpm for Philips, and 68 ± 6 bpm for Siemens images. LS assessments were feasible in 1188 segments with QLAB/aCMQ, 1214 segments in QLAB/LV auto-strain, 1225 segments in VVI, 1234 segments in EchoPAC and 1257 segments in EchoInsight. In the aggregate mean of all software packages, absolute global longitudinal strain values were lowest in magnitude for cardiac amyloidosis at -12.3 ± 3.8%, followed by HCM -17.6 ± 2.4%, and highest for controls -21.0 ± 1.4%. MDI measurements were lowest for controls at 38 ± 7 ms and similarly higher for HCM 65 ± 15 ms and cardiac amyloidosis 64 ± 20 ms (p<0.001). **Fig 3** shows that PSI

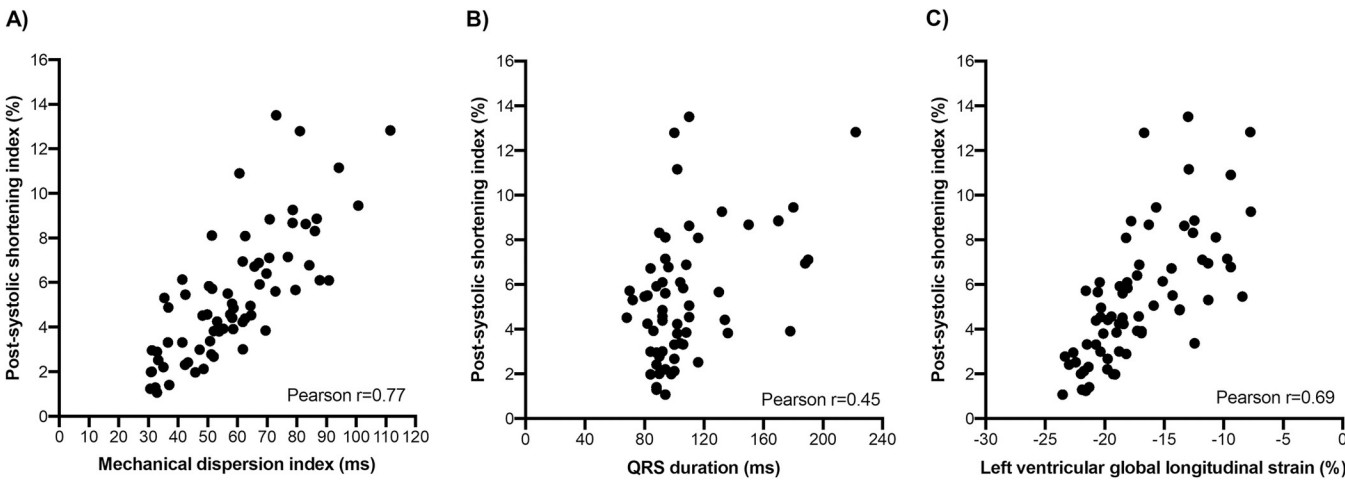

**Fig 5.** Correlation scatterplots between post-systolic shortening index and (a) mechanical dispersion index, (b) QRS duration and (c) left ventricular global longitudinal strain.

measurements were also lowest in controls at 2.1 ± 0.6%, and higher for HCM at 6.1 ± 2.6% and cardiac amyloidosis at 6.8 ± 2.7% (p<0.001).

### Variability of PSI measurements among software vendors

The results of PSI measurements by various software vendors in all subjects are shown in **Fig 4A**, with mean PSI values varied significant from 3.7% to 6.2% (p<0.001). In the control group, mean PSI among software vendors ranged from 1.4% to 2.9%, with no significant differences observed (p = 0.11, **Fig 4B**). PSI among software vendors varied significantly from 4.7% to 7.2% for HCM patients (p = 0.003, **Fig 4C**), and also varied significantly from 3.8% to 8.5% for cardiac amyloidosis patients (p = 0.008, **Fig 4D**).

### Correlations between PSI and other strain and dyssynchrony variables

**Fig 5** displays the scatterplots and Pearson correlation coefficients between averaged measurements across all five software vendors of PSI and (a) MDI, (b) QRS duration and (c) LVGLS. There were moderate to high correlations between PSI and MDI (r = 0.77) and PSI and LVLGS (r = 0.69), and modest correlations between PSI and QRS duration (r = 0.45). In receiver-operative characteristic analyses to identify left bundle branch block, MDI had area under curve of 0.87 (p = 0.002, optimal threshold 63ms giving sensitivity 100% and specificity 70%), while PSI had area under curve of 0.90 (optimal threshold 7.6% giving sensitivity 86% and specificity 87%). For identifying QRS duration ≥120 ms, MDI had area under curve of 0.79 for detecting left bundle branch block (p = 0.001, optimal threshold 58ms giving sensitivity 92% and specificity 61%), while PSI had area under curve of 0.80 for detecting left bundle branch block (optimal threshold 6.9% giving sensitivity 69% and specificity 82%).

### Discussion

Our data extend previous findings [9] demonstrating that PSI occurs in a variety of LV pathologies, and is not just a manifestation of acute or chronic coronary artery disease. The correlation between PSS and cardiac pathology may be a reason why it can act as survival predictor in the general population [10]. We also show that changes in PSI in general follow the changes in

MDI, which suggests that PSS is influenced by dyssynchrony that is intrinsic to the presence of concentric LV hypertrophy [4]. Finally, values of PSI in individual patients have to be interpreted while understanding that there is considerable bias and lack of agreement among software vendors. The bias we detected was not constant: it was absent in controls, but significant in patients with pathologically increased LV wall thickness.

## Speckle-tracking echocardiography derived parameters as indices of dyssynchrony

While numerous echocardiographic parameters have been proposed as the optimal means to evaluate LV dyssynchrony, this study highlights their inter-relationship. One such paraeter MDI, which, while was originally proposed as a predictor of sudden cardiac death in ischemic and non-ischemic cardiomyopathies [11–15], has more recently been shown to be useful in risk stratification of patients with HCM [16–19]. Increases in MDI in HCM is likely related to hypertrophy-associated fibrosis of cardiac muscle leading to electrical inhomogeneity of intraventricular conduction. While MDI assesses the temporal strain variability caused by dyssynchronous LV contraction, PSI measures the amplitude variability of strain following aortic valve closure [20]. Although PSI is commonly associated with myocardial ischemia and has been proposed as sensitive marker of regional dysfunction, it can also be observed in various other settings. Voigt et al. found that 30% of healthy individuals exhibited some degree of PSI [21]. The results from the present study highlight that PSI is frequent in patients with pathologic increase in LV wall thickness. Of note, both MDI and PSI had significant correlations with QRS duration, indicative of their relationship of LV dyssynchrony.

Usually, electromechanical dyssynchrony and regional LV dysfunction brought by perfusion abnormality are thought of as separate mechanisms with different "phenotypes". Yet, the two processes of dyssynchrony and hypoperfusion can coexist and reinforce each other. Lionetti et al. showed that, in an animal model, LV dyssynchrony induced by pacing leads to hibernation [22]. Prior studies have also identified PSI as a potential imaging biomarker for ischemia [2]. We provide further proof of intrinsic link between temporal parameters of regional dyssynchrony such as MDI and parameters of amplitude of segmental wall contraction such as PSI, and these are accentuated in patients with pathological increase in left ventricular wall thickness.

## Variability of speckle-tracking echocardiography derived parameters among software

We show that PSI measurements vary considerably among software vendors. The cause of the bias in PSI measurements, and why it appears in subjects with increased wall thickness, is likely multifactorial. One of the factors is the way the individual strain algorithms work. **Fig 1** illustrates this point. There are perceptible between-software differences in the shape of global and segmental strain curves. The varying amount and ease of user interaction with measurements may also affect these variations in MDI and PSI measurements. Another reason could be the way measurements of PSI and GLS interact [12]. The dispersion of differences in MDI and PSI measurements among software packages were larger in patients with increased LV thickness (and decreased GLS) compared with controls. These findings mean that the patients with low absolute GLS value may have more variation in time-to-peak measurements than patients with high absolute GLS value simply because of a better separation between signal and noise on the high-amplitude strain curve. This may also have clinical significance, as it implies that subjects with lower systolic function may have larger MDI despite not always having larger dyssynchrony.

## Clinical implication

This is the first study to report about the presence of PSI in patients with HCM and amyloidosis, with the diagnostic and prognostic value of PSI not previously assessed in these diseases. Our study shows that mean PSI value of all software products in the setting of these populations significantly increased compared with controls. PSI and MDI in patients with HCM and amyloidosis are potentially useful for assessing the risk stratification of ventricular arrhythmias, as it seems that shortening after aortic valve closure reflects segmental myocardium dysfunction or electrical conduction dysfunction. Further studies are warranted to assess the association of increased PSI with ventricular arrhythmia and sudden death events in these populations. If these associations exist, PSI and/or MDI should be routinely assessed in these patients when echocardiography is performed, and if abnormally elevated then more frequent heart rhythm monitoring may be considered. It appears clinically important that when one evaluates MDI and PSI, cut-off values for abnormal MD and PSI derived from speckle tracing echocardiography should be software specific. Another area of potential use of PSI is assessment in "grey zone populations" such as left ventricular hypertrophy in athletes with no clear diagnosis of cardiomyopathy [23]. Distinguishing hypertrophic cardiomyopathy from athlete's heart represents a clinical problem of increasing magnitude and significance [24, 25]. While no data are available on the impact of athlete's heart on PSI or MDI, one can speculate that these patients have preserved long axis function, and PSI would be less abnormal than HCM. Further studies are needed to assess this issue.

## Limitation

This study should be interpreted in the context of the following limitations. First, we limited our research to only 5 out of at least 9 commercially available longitudinal strain software types due to licensing availability. However, the findings are generalizable in that software packages cannot be assumed to be interchangeable unless they are specifically tested. The second is that, while our HCM and amyloid patients often showed a severely abnormal GLS, we have not tested how different software packages perform in subjects with thin LV walls and abnormal GLS. While the expectation would be that thinner walls would increase noise, and thus increase the values of PSI, this is an unproven hypothesis. Still, one can expect that the problem with bias and the limits of agreement would not be eliminated with decreasing LV wall thickness. Thirdly, the sample size of this prospective study limits the power for analyses. Lastly, longer-term outcomes were not collected to allow for prognostic evaluation of PSI.

## Conclusion

PSI was more elevated in patients with cardiac amyloidosis and HCM than controls. While there were significant correlations in PSI measurements among echocardiographic software vendors, the confidence interval for the agreement in PSI measurements among vendors were large. Cut-off values for abnormal PSI should be software specific. Furthermore, we demonstrate that there was a significant difference in PSI measurements between software vendors in patients with pathological increase in LV wall thickness, while comparable in controls. PSI also correlates with other echocardiographic parameters, supporting its use in mechanical dyssynchrony assessments.

## Supporting information

**S1 Table. Minimal dataset of this prospective cohort study.**
(DOCX)

## Author Contributions

**Conceptualization:** Yoshihito Saijo, Tom Kai Ming Wang, Brett W. Sperry, Dermot Phelan, Milind Y. Desai, Brian Griffin, Richard A. Grimm, Zoran B. Popović.

**Data curation:** Yoshihito Saijo, Tom Kai Ming Wang, Nicholas Chan, Brett W. Sperry, Dermot Phelan, Zoran B. Popović.

**Formal analysis:** Yoshihito Saijo, Tom Kai Ming Wang, Nicholas Chan, Zoran B. Popović.

**Investigation:** Yoshihito Saijo, Tom Kai Ming Wang, Nicholas Chan, Zoran B. Popović.

**Methodology:** Yoshihito Saijo, Tom Kai Ming Wang, Nicholas Chan, Zoran B. Popović.

**Project administration:** Zoran B. Popović.

**Resources:** Zoran B. Popović.

**Supervision:** Zoran B. Popović.

**Validation:** Tom Kai Ming Wang, Zoran B. Popović.

**Writing – original draft:** Yoshihito Saijo, Tom Kai Ming Wang, Zoran B. Popović.

**Writing – review & editing:** Yoshihito Saijo, Tom Kai Ming Wang, Nicholas Chan, Brett W. Sperry, Dermot Phelan, Milind Y. Desai, Brian Griffin, Richard A. Grimm, Zoran B. Popović.

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
