## [Decision Letter · Decision Letter 0]

1 Jul 2022

PONE-D-22-14707Post-systolic shortening index by echocardiography evaluation of dyssynchrony in the non-dilated and hypertrophied left ventriclePLOS ONE

Dear Dr. Popović,

Thank you for submitting your manuscript to PLOS ONE. After careful consideration, we feel that it has merit but does not fully meet PLOS ONE’s publication criteria as it currently stands. Therefore, we invite you to submit a revised version of the manuscript that addresses the points raised during the review process.

ACADEMIC EDITOR: The authors should more focus on the clinical implications and perspectives. Moreover, they should discuss the role of regional left ventricular dyssynchrony related to potential myocardila hybernation in the light of previous unmentioned pre-clinical study (please see J Card Fail. 2009 Dec;15(10):920-8).

We look forward to receiving your revised manuscript.

Kind regards,

Vincenzo Lionetti, M.D., PhD

Academic Editor

PLOS ONE

Journal Requirements:

Reviewers' comments:

Reviewer's Responses to Questions

**Comments to the Author**

1. Is the manuscript technically sound, and do the data support the conclusions?

Reviewer #1: Yes

2. Has the statistical analysis been performed appropriately and rigorously? 

Reviewer #1: Yes

3. Have the authors made all data underlying the findings in their manuscript fully available?

Reviewer #1: Yes

4. Is the manuscript presented in an intelligible fashion and written in standard English?

Reviewer #1: Yes

5. Review Comments to the Author

Reviewer #1: This is the first study to report about the presence of PSI in patients with HCM and amyloidosis, showing that mean PSI value of all software products in the setting of these population significantly increased compared with controls. Congratulation to the authors, here you find some comments in order to improve the manuscript

Discussion :It is clear how patients with HCM may have more pathologic post-systolic shortening, which may have etiologic contribution to the functional heterogeneity of this disease entity especially diastolic dysfunction (DOI: 10.1016/j.echo.2006.03.019). In this scenario authors should also focus more on the considered HCM and amyloidosis diagnostic criteria. Please add a nice figure

In discussion please explain author’s thoughs about post-systolic shortening in “grey zones population” such subjects with athlete’s heart but no clear diagnosis of cardiomyopathy (please cite: DOI: 10.1111/jce.14526). In fact, today distinguishing hypertrophic cardiomyopathy from athlete’s heart represents a clinical problem of increasing magnitude and significance (doi: 10.1136/hrt.2005.060962; DOI: 10.1016/j.ijcard.2021.10.013). Is PSS minimally influenced by dyssynchrony that is intrinsic to the presence of physiologic LV hypertrophy? Please discuss and cite 4 suggested references

It is definitely not clear how many patients underwent to CMR (providing best comprehensive characterization of the HCM, cite DOI: 10.1016/j.jcmg.2019.09.020) to genetic testing (DOI: 10.3390/ijms221910401) . Moreover how many patients were symptomatic? (the importance of syncopal diagnosis due to LVOT obstruction), did you find a difference in your patient population? Please explain in the discussion/limitations the importance of these topics, why you did not investigate and cite all suggested references

In the paragraph “Clinical implication” more author’s considerations among MDI and PSI as potentially useful for assessing the risk stratification of ventricular arrhythmias, are welcome. Moreover how would you follow these patients? Holter? Loop recorder? Only symptoms? Please explain

6. PLOS authors have the option to publish the peer review history of their article (what does this mean?). If published, this will include your full peer review and any attached files.

Reviewer #1: No

---

## [Author Response · Author response to Decision Letter 0]

23 Jul 2022

See file attachment for responses to reviewers' comments. Please note that we are providing images/graphs in a response.

---

## [Decision Letter · Decision Letter 1]

9 Aug 2022

Post-systolic shortening index by echocardiography evaluation of dyssynchrony in the non-dilated and hypertrophied left ventricle

PONE-D-22-14707R1

Dear Dr. Popović,

We’re pleased to inform you that your manuscript has been judged scientifically suitable for publication and will be formally accepted for publication once it meets all outstanding technical requirements.

Kind regards,

Vincenzo Lionetti, M.D., PhD

Academic Editor

PLOS ONE

Additional Editor Comments (optional):

Reviewers' comments:

Reviewer's Responses to Questions

**Comments to the Author**

1. If the authors have adequately addressed your comments raised in a previous round of review and you feel that this manuscript is now acceptable for publication, you may indicate that here to bypass the “Comments to the Author” section, enter your conflict of interest statement in the “Confidential to Editor” section, and submit your "Accept" recommendation.

Reviewer #1: All comments have been addressed

2. Is the manuscript technically sound, and do the data support the conclusions?

Reviewer #1: Yes

3. Has the statistical analysis been performed appropriately and rigorously? 

Reviewer #1: Yes

4. Have the authors made all data underlying the findings in their manuscript fully available?

Reviewer #1: Yes

5. Is the manuscript presented in an intelligible fashion and written in standard English?

Reviewer #1: Yes

6. Review Comments to the Author

Reviewer #1: Authors performed good answers to all reviewer's suggestions. Congratulations, manuscript definitely improved

7. PLOS authors have the option to publish the peer review history of their article (what does this mean?). If published, this will include your full peer review and any attached files.

Reviewer #1: No

---

## [Editor Report · Acceptance letter]

16 Aug 2022

PONE-D-22-14707R1 

Post-systolic shortening index by echocardiography evaluation of dyssynchrony in the non-dilated and hypertrophied left ventricle 

Dear Dr. Popović:

I'm pleased to inform you that your manuscript has been deemed suitable for publication in PLOS ONE. Congratulations! Your manuscript is now with our production department. 

Kind regards, 

on behalf of

Prof. Vincenzo Lionetti 

Academic Editor

PLOS ONE